# Virtual Reality and Augmented Reality in Plastic and Craniomaxillofacial Surgery: A Scoping Review

**DOI:** 10.3390/bioengineering10040480

**Published:** 2023-04-17

**Authors:** Nicolas Kaplan, Mitchell Marques, Isabel Scharf, Kevin Yang, Lee Alkureishi, Chad Purnell, Pravin Patel, Linping Zhao

**Affiliations:** 1Division of Plastic, Reconstructive and Cosmetic Surgery, College of Medicine, University of Illinois at Chicago, Chicago, IL 60612, USA; nkapla3@uic.edu (N.K.); mmarqu35@uic.edu (M.M.); ischarf2@uic.edu (I.S.); 2The Craniofacial Center, Division of Plastic, Reconstructive, and Cosmetic Surgery, University of Illinois at Chicago, Chicago, IL 60612, USA; keviny.umd@gmail.com (K.Y.); alkureishi.lee@gmail.com (L.A.); cpurnell@uic.edu (C.P.); pkpatel@uic.edu (P.P.); 3Shriners Children’s Chicago Hospital, Chicago, IL 60707, USA

**Keywords:** virtual reality, augmented reality, plastic surgery, craniofacial surgery, surgical planning

## Abstract

Virtual reality (VR) and augmented reality (AR) have evolved since their introduction to medicine in the 1990s. More powerful software, the miniaturization of hardware, and greater accessibility and affordability enabled novel applications of such virtual tools in surgical practice. This scoping review aims to conduct a comprehensive analysis of the literature by including all articles between 2018 and 2021 pertaining to VR and AR and their use by plastic and craniofacial surgeons in a clinician-as-user, patient-specific manner. From the initial 1637 articles, 10 were eligible for final review. These discussed a variety of clinical applications: perforator flaps reconstruction, mastectomy reconstruction, lymphovenous anastomosis, metopic craniosynostosis, dermal filler injection, auricular reconstruction, facial vascularized composite allotransplantation, and facial artery mapping. More than half (60%) involved VR/AR use intraoperatively with the remainder (40%) examining preoperative use. The hardware used predominantly comprised HoloLens (40%) and smartphones (40%). In total, 9/10 Studies utilized an AR platform. This review found consensus that VR/AR in plastic and craniomaxillofacial surgery has been used to enhance surgeons’ knowledge of patient-specific anatomy and potentially facilitated decreased intraoperative time via preoperative planning. However, further outcome-focused research is required to better establish the usability of this technology in everyday practice.

## 1. Introduction

Recent advances in imaging technology have greatly benefitted the field of medicine by allowing a better 3D visualization of anatomy in settings such as medical education and surgical planning. In particular, virtual reality (VR) and augmented reality (AR) have been applied to the medical field since the 1990s, although their adoption was limited by the quality of experience [1,2,3,4]. These technologies primarily strive to bridge the gap between two-dimensional imaging modalities and the three-dimensional nature of surgical procedures [5]. Furthermore, multimodal imaging is required to formulate a differential diagnosis in tumor surgery applications or vascular malformations by providing diagnostic evidence on soft tissues [6]. Computer-aided design software allows for the manipulation of patient anatomical data, yet it is limited in usability by its technical complexity and often prohibitive cost [2,4,7,8].

Early VR technology was hindered by the issue of latency—that is, the delay between when the user moves their head and when the virtual image is adjusted—provoking vestibulocochlear vertigo in the user. It was not until a series of technological developments in the early 2010s—many driven by the parallel development of smartphones—that AR/VR technology became more widespread. Amongst these improvements were the refinement of the “positional tracking” of a user’s eyes to direct camera orientation, the enhancement of high-definition screen capabilities, and rotational tracking maturation via magnetic and computer-based methods [4,9,10,11]. The issue of vestibulocochlear vertigo was largely resolved by restricting VR screen rotation to only occur alongside the concurrent rotation of the head, minimizing the disconnection of perceived motion between the eyes and inner ear [4]. These innovations, alongside improvements in computer processing capabilities and the miniaturization of VR and AR hardware, set the stage for commercially accessible VR/AR platforms. Among the first were the Oculus Rift (Oculus VR, Menlo Park, CA, USA) and HoloLens (Microsoft Corporation, Redmond, WA), both of which were released in 2016 [4]. Commercial VR/AR platforms have proven to be fertile ground for video game companies, and these companies have built standardized software platforms for VR/AR systems; some, such as Unreal Engine (Epic Games, Cary, NC, USA) and Unity (Unity Technologies, San Francisco, CA, USA), have become the basis for numerous other software products [4,12,13,14,15,16].

Novel VR and AR platforms have proven to be a boon in the medical field, particularly where anatomic visualization is desired, but fully accomplishing anatomic visualization in the operating room is difficult. Studies on the use of VR/AR in surgery have identified several potential benefits: intuitive viewing of patient-specific anatomy, estimation of outcomes via pre-surgical planning, reduced intraoperative time, and decreased complications when VR/AR is used for intraoperative navigation [4,5]. Other recorded benefits of VR/AR include its relative affordability: The technology, which can utilize consumer-grade hardware, is a cost-effective resource that can enhance operative plans while potentially reducing associated intraoperative time and cost [7,17]. Beyond the operating room, VR/AR offers a platform for both patient and physician education. VR has already been successfully tested in preoperative education for trainees and as a simulator prior to attempting surgery [18,19]. For instance, the development of virtual dissection tables, which are interactive devices that provide male and female cadaver datasets by different modes (gross anatomy and high-resolution methods), is a promising key element in anatomical teaching [20]. For patients, the use of VR to explain prospective procedures has been found to potentially decrease preoperative anxiety [21,22,23]. Typically, educational and patient-as-user applications of VR and AR do not require a strict timeline. In contrast, in clinician-as-user settings, the timeliness of VR or AR generation is important to facilitate the efficiency of the surgical procedure.

Existing reviews have broadly surveyed the use of VR/AR in the field of medicine, including in the domains of surgical training, education, planning, and navigation [4,22]. To date, thousands of healthcare-related VR articles have been published [4,24]. Moreover, nearly USD 14 billion was expected to have been spent on VR and AR globally in 2022, with growth expected to eclipse USD 50 billion by 2026 [25].

Innovations in VR and AR software and hardware have broadened accessibility to these platforms and made them more widespread than ever. In addition to novel medical applications that have begun to utilize mainstream devices such as the HoloLens and software platforms based on major software templates, powerful new smartphones have arisen as a growing platform for AR both within and outside of the clinical setting [12,13,14,15,16,26,27,28,29,30]. VR/AR also has the capacity to synergize with existing “high tech” surgical aids such as 3D printing technology [24,31].

The intersection of these technological advances, commercial accessibility, investment capital, and the ability to integrate into current practices in the mid-2010s have made understanding the present scope of VR and AR more important than ever. The field of plastic and craniomaxillofacial surgery is well positioned to explore these benefits since the use of three-dimensional imaging and patient-specific planning to create customized cutting guides, and prosthetics is a well established adjunct to a variety of procedures. We therefore aimed to conduct a scoping review of the use of VR and AR in the specific contexts of preoperative surgical planning and intraoperative navigation in order to elucidate their impact on patient care and individualized surgical procedures.

## 2. Materials and Methods

### 2.1. Protocol and Registration

The literature search and writing of this scoping review were conducted in accordance with the Preferred Reporting Items for Systematic Reviews and Meta-Analyses extension for Scoping Reviews (PRISMA-ScR) guidelines. This is a scoping review, a relatively novel study design alternative to systematic reviews in clinical research. As Munn et al. 2018 described, a scoping review is used to “identify knowledge gaps, scope of a body of literature, clarify concepts or to investigate research conduct [32]”.

### 2.2. Eligibility Criteria

Publications were included if they were (1) within the scope of plastic and craniomaxillofacial surgery, (2) published between 2015 and 2022, (3) written in English, (4) used AR or VR for a patient-specific application, involved human participants, and the “user” of the technology was the clinician performing or rehearsing the procedure (i.e., “clinician-as-user” applications).

Exclusion criteria were (1) VR or AR utilization for medical education, training, or performance assessment; (2) non-English, non-full-text articles published before 2015; (3) articles out of scope of plastic and craniomaxillofacial surgery; (4) integrated AR-robot assisted applications (e.g., laparoscopic and neuronavigation); (5) “patient-as-user” applications (e.g., patient education); and (6) studies that generally did not implement AR/VR for a patient-specific purpose.

### 2.3. Search

A literature search was conducted on August 6 2021 by MG and repeated and updated on November 2 2022 by NK utilizing the PubMed, Embase, and Ovid Medline databases. The following keywords and associated Boolean operators were used: (virtual reality OR “augmented virtuality” OR augmented reality OR “mixed reality” OR “extended reality”) AND (plastic surgery OR cosmetic surgery OR reconstructive surgery OR aesthetic surgery OR maxillofacial surgery OR orthognathic surgery) AND (“patient-specific” OR customiz* OR individual* OR tailor* OR person*).

### 2.4. Selection Criteria

Search results were uploaded to Covidence (Veritas Health Innovation, Melbourne, Australia) and assessed by two independent authors (MM, NK) in a two-step process according to a priori screening standards as described in the eligibility criteria. Articles were first screened by title and abstract; if it was unclear whether the study should be included based on the title and abstract alone, the entire article was assessed. After an initial assessment, the eligible full-text articles were read by two independent authors. If the two authors disagreed about inclusion or exclusion, a third author read the article and ruled for a consensus.

### 2.5. Data Collection Process

A data collection form within Covidence (Veritas Health Innovation, Melbourne, Australia) was jointly developed by two reviewers (MM, NK) to determine which variables to extract. The two reviewers independently charted the data, discussed the results, and continuously updated the form.

### 2.6. Data Items

Data were extracted on demographics (e.g., country of origin, institution), specific aims and conclusions, technology characteristics (e.g., AR or VR, device(s), and software), surgical procedure, the timing of implementation, study design, and participant characteristics.

## 3. Results

### 3.1. Search Results

The literature search identified 1637 studies, with 1029 citations remaining after duplicates were removed. Based on the titles and abstracts, 823 were excluded. In total, 196 of the remaining 206 full-text articles were excluded for the following reasons: 103 were not in the field of plastic surgery, 54 were either non-English or not full-text articles (e.g., abstracts and conference papers), 22 focused on robot-assisted surgery, 9 focused on medical training or education, 6 were not patient-specific, 1 was related to patient education, and 1 was a review article. The remaining 10 articles were considered eligible for this review. See Figure 1 for the full search protocol.

### 3.2. Characteristics of Included Studies

Table 1 depicts a summary of key points derived from each included article in line with the scope of this review. Subsequent sections elaborate on the information shown in Table 1.

### 3.3. Year of Publication

The 10 included articles were distributed between 2018 and 2022. In total, one of these was published in 2018, two were published in 2019, three were published in 2020 and 2021, respectively, and one was published in 2022. 

### 3.4. Type of Paper

All included studies described a workflow for the implementation of a distinct AR or VR platform in the context of a given surgical procedure in a clinician-as-user, patient-specific manner. Three studies elaborated on their workflow description by examining the application and/or outcomes in a case report. 

### 3.5. Timing of Implementation

The actual implementation of VR/AR was split between preoperative and intraoperative use. In total, 6/10 of these used AR/VR intraoperatively, and the remaining used AR/VR preoperatively.

### 3.6. Clinical Context

VR and AR were used in the context of the following procedures: perforator flaps reconstruction, mastectomy reconstruction, lymphovenous anastomosis, metopic craniosynostosis, dermal filler injection, auricular reconstruction, facial vascularized composite allotransplantation, and facial artery mapping. Of these, AR was studied in the context of deep inferior epigastric perforator (DIEP) in two studies and metopic craniosynostosis repair in two more. Both DIEP papers involved the intraoperative use of AR, whereas metopic craniosynostosis articles were split between preoperative and intraoperative use. Of note, the investigation of AR use in facial artery mapping had a secondary aim of proving its capacity to be used in dermal filler injections; as a result, this review considers dermal filler injections as the relevant clinical context in Waked et al. for inclusion in this scoping review.

### 3.7. Hardware

In total, 4/10 studies involved Microsoft HoloLens, and another 4 used smartphones as the platform for the VR and AR applications. Smartphone-based applications used varying brands: Android only [13], iOS only [29], both Android and iOS [14], or not listed [28]. The remaining two studies used a Vive Pro VR headset (HTC Corporation, New Taipei City, China) and Epson Moverio BT-300 AR glasses (Epson, Suwa Japan). Overall, 9/10 studies implementing AR solutions utilized HoloLens, smartphones, and Epson BT-300. Only Vive Pro was utilized for a VR study.

### 3.8. Software

In total, 3/10 studies did not report the software used for their VR or AR applications. For those that reported the software type, 6/10 used some version of the Unity Gaming platform (Unity Technologies, San Francisco, CA, USA) as a basis for their custom-built software. The remaining study used Medicalholodeck (Zurich, Switzerland), a software for visualizing and manipulating 3D CT data in the form of DICOM files.

## 4. Discussion

### 4.1. Three-Dimensional Imaging in Plastic and Craniomaxillofacial Surgery

Three-dimensional modeling has become the cornerstone for preoperative planning and the production of intraoperative surgical aids. Presurgical planning in particular has become an important tool for planning and executing intricate procedures within the scope of plastic and craniomaxillofacial surgery [33]. Such modeling can take several forms, including 3D computer planning, 3D printing, and augmented and virtual reality. More traditional computer-based planning has been seen to improve accuracy, efficiency, and reproducibility within plastic surgery [14,34,35]. Three-dimensionally printed models, based on patient CT scans, have built upon computer planning and demonstrated their capacity to assist in plastic surgery and other surgical fields as preoperative and intraoperative tools [5,13,27,30,36,37]. VR and AR have arisen as potential alternatives or adjuncts to existing modalities to allow for more comprehensive and intuitive surgical planning. Some studies have created hybrid models, integrating AR technology with pre-existing 3D-printed models for more comprehensive surgical planning [13]. Others have compared 3D printing to AR, noting how the color palette and material chosen for printing can distort printed models relative to AR ones—despite the fact that 3D printing remains a gold standard in surgical planning [27]. Regardless of these comparisons, VR and AR have, respectively, demonstrated the ability to enhance the surgeons’ operative view via the real-time visualization of patient-specific anatomy pre- and intraoperatively [15,16,26,38]. The ability to interact with patient-specific anatomy in three dimensions can be particularly useful during complex plastic surgery procedures [15,26,27,30].

### 4.2. Methodologies of the Included Studies

A variety of methodologies were implemented among the included studies, with an overarching pattern of the use of a VR or AR platform in a case or series of cases. Regardless of the procedure of use and time of implementation, 3D imaging was initially conducted for transference to AR/VR platforms. The only exception to this was Amini et al., who did not need to visualize participant anatomy as they instead projected the 3D presence of a breast implant for mastectomy reconstruction patients [12]. The majority of studies paired their VR or AR application with a model; Cho et al., Yaremenko et al., Coelho et al., and Garcia-Mato et al. utilized 3D-printed, patient-specific models parallel to their virtual platform [13,14,27,30]. The accuracy of the virtual platforms was validated to varying degrees. Cho et al. compared their AR hologram to 3D CT data, while Amini et al. compared theirs to an inflatable model [12,27]. Garcia-Mato overlaid their AR model onto a 3D-printed model [14]. Ultrasound was used to validate accuracy in studies wherein AR was used to identify vasculature [16,30]. In the single VR study, the preoperative planning of lymphaticovenous anastomosis was checked intraoperatively using indocyanine green lymphography [26]. Notably, while AR platforms can be utilized either pre- or intraoperatively, VR applications are primarily used as planning tools preoperatively.

### 4.3. Preoperative and Intraoperative Use in Plastic and Craniomaxillofacial Surgery

Perhaps most importantly, VR/AR was noted as a tool to potentially improve surgical outcomes. VR and AR may facilitate different perspectives of patient anatomy while planning and operating. Patient-specific modeling can optimize intraoperative actions. For example, in the context of LVA, AR facilitated preoperative planning and thereby minimized iatrogenic damage and scarring [26]. Preoperative AR planning further enabled the accurate intraoperative reproduction of practice in craniosynostosis repair [13]. One unique application of AR was the development of a “sharing” function, allowing for multiple surgeons to work on a preoperative model simultaneously [27]. AR was found to increase accuracy in planning, providing accurate models for even the most anatomically complex craniofacial/plastic surgery procedures with the added benefits of reduced costs and saved time [27]. Such features could be utilized by users in distinct locations, enabling presurgical collaboration across time and space [27,39].

Intraoperative AR-guided procedures were also determined to be accurate by subsequent Doppler ultrasound comparisons [15]. Similarly, the implementation of AR in auricular reconstruction was noted to streamline an initial stage of the procedure and reduce the risk of intraoperative injury [30,40]. AR was also seen to be useful for minimizing risks of complications with routine procedures such as filler administration [28,29]. AR allows the surgeon to directly overlay the planned interventions onto patient anatomy, thus minimizing deviations from the plan while still allowing the surgeon to make any necessary alterations intraoperatively to optimize patient outcomes [14]. AR models can provide different layers of holographic projections (e.g., blood supply and muscle) and distinct color schemes for distinguishing various tissues [15,27]. By registering the AR hologram to the surgical field, the projection may be viewable in full—or at varying degrees of complexity—throughout the course of an operation [14,16]. However, different VR/AR products had varying success in accomplishing continued hologram projection as tissue deformation occurred; this was noted as a major concern or aim for future improvement [12,15]. 

Sterility was not seen as an issue during the intraoperative implementation of AR. Head-mounted devices function independently of computers and are self-contained—utilizing vocal inputs and hand gestures registered by sensors for interaction with holograms [12,15,27]. Moreover, any markers or other tools used to register the hologram to the patients can be safely sterilized as well [16]. Even when using handheld devices such as smartphones as an AR platform, sterility can be maintained via sterile phone or ultrasound probe covers, which do not interfere with screen interactions [14]. Moreover, the same digital rendering and AR hologram may be transferable between various hardware platforms. For example, when users found the HoloLens headset to be cumbersome in the study by Amini et al., the authors suggested shifting the subsequent iteration of their platform to a smartphone application.

Unfortunately, there is a scarcity of accurate data in the literature. While some studies have carried out preliminary accuracy analysis by comparing frontal and lateral cephalograms prior to and after surgery, they acknowledge the need for more complete accuracy analysis using tools such as 3D CT scans [41]. As AR and VR platforms continue to integrate into clinical practice, there is hope that further accurate data will be reported [14,15,27,28,29,41]. The existing literature has nevertheless demonstrated improved safety and replicability for surgical procedures that utilize AR platforms [13,14,15,41]. The improved knowledge of patient-specific anatomy by using AR and VR may allow for faster and more confident identification of the surgical site, thus potentially contributing to increased safety via reduced anesthetic and intraoperative duration [15,27]. 

### 4.4. Broader Utility of VR and AR in Surgery 

The technology behind VR and AR implementation is itself flexible, as evidenced by the wide variety of hardware interfaces employed by the studies described here. These varied implementations allow the technology to be effectively used both preoperatively and intraoperatively as the need arises. 

A benefit of the aforementioned flexibility has been demonstrated by the use of VR and AR in multiple other surgical specialties. One AR system as early as 1997 utilized a mirror at an angle to create a parallax view with overlaid patient CT data, assisting neurosurgeons in removing glioblastomas [42]. 

AR has also been implemented in general surgery since as early as 2004, when a 3D rendering of an adrenal adenoma and surrounding abdominal organs was overlaid on the laparoscopic camera screen during adrenalectomy [43]. The use of AR/VR is well documented with respect to the surgical approach, and such technology has been further applied in the context of otolaryngologic procedures to provide real-time guidance prompted by intraoperative images or video feeds [44]. For neurosurgery, VR has been effectively implemented in pre-surgical planning for tumor resection, with demonstrated superior surgical outcomes in sellar region surgeries as well as skull base tumor resections [45,46]. Orthopedic and spinal applications include the development of presurgical plans in VR that can be uploaded for surgical navigation as well as AR-based applications for the placement of sacroiliac screws, the fixation of cervical fractures, and bone tumor resections [47,48,49].

Beyond clinical use, VR has been extensively studied in its role as a surgical training tool in multiple fields [50,51,52,53]. VR was shown to be a viable introductory tool for novice trainees that potentially eased the learning curve for surgical procedures in the context of laparoscopic surgery [50]. Indeed, VR simulators exist for a variety of techniques, including but not limited to robotic systems and surgical procedures such as those in cardiothoracic and orthopedic surgery [51].

Notable, in line with the results of this scoping review is the relative scarcity of validation data for VR and AR alike: Multiple systematic reviews spanning specialties such as ophthalmology, neurosurgery, and orthopedics found absent or weak quantitative evidence for the implementation of such devices into practice [54]. One recent scoping review examined the role of VR and AR in a clinician-as-user, patient-specific manner, albeit with broader criteria than in this review: It included studies that examined any surgical application of any specialty, including those examining applications in medical education as well as experiments conducted with cadavers. Consistent with the findings of this review, the implementation of extended reality was reported to contribute to the surgical field by enhancing intraoperative spatial awareness and reducing the risk of iatrogenic injury [55]. In further alignment with this review’s findings, the authors concluded that there is a significant need for further studies—particularly requiring more data reported with respect to the quantification of the accuracy of operative plans and outcomes using these extended reality applications [55].

### 4.5. Cost and Time Savings

Beyond the direct impact of assisting a surgeon in conducting an optimized procedure, patient-specific anatomic modeling via AR was observed to have more mixed impacts on operative time and costs. For example, one novel application of AR for orthognathic surgery extended operative times by one hour due to issues with registering the hologram to the patient and handling the device [41]. However, in such cases, participants indicated that further training would help reduce any delays [14,41]. Other studies demonstrated the opposite, with operative times being decreased compared to gold standards in their respective surgery [15,27]. Pratt et al. found their approach to be more consistent than ultrasound, which itself was limited in its ability to accurately distinguish the relationship between arteries and musculature at greater depths and was seen as inferior to alternatives such as CTA for the purpose of arterial identification in perforator flap surgery [15,16,56,57,58,59]. This was considered to result in the increased efficiency of their AR platform [15]. In the case of facial vascularized composite allografts, 3D printing donor and recipient models took a cumulative 47 h vs. 14 h of rendering AR holograms [27]. 

Similarly to mixed findings on the impact of VR and AR on time, different findings on the impact of the technology on cost were present. When considering the use of MRA to minimize risks in dermal filler injection, for example, a cost of approximately EUR 250 for an MRA in Western Europe was attributed to the methodology [28]. Yet, Mespreuve et al. felt the price to be acceptable when compared to yearly filler injection costs and the risk of blindness [28]. While few studies discussed the cost of the VR/AR technology itself, virtual simulation was noted to be a cost-saving mechanism by reducing time, modeling, and errors [7,17,27]. One study reported a 100-fold decrease in cost compared to 3D printing for the purpose of operative planning [27].

Overall, further quantification of the savings enabled by implementing these technologies is needed. The majority of the articles included in this review primarily focused on the development and implementation of novel workflows using these technologies. As a result, they did not comprehensively delineate nor compare the effect of their platform on surgical costs and timing. There is a need for further evaluation of these variables.

### 4.6. Limitations

#### 4.6.1. Data Collection

One limitation of this scoping review is the constrained timeline from Jan 1 2015 onwards. The innovations in both hardware and software, which occurred in late 2015 and 2016, were the impetus for selecting the search’s start date; however, this could have limited final results via the involuntary exclusion of any relevant studies published before 2015 [4]. There is also potential for the incomplete retrieval and identification of articles in the literature pertaining to the utilization of AR/VR in plastic and craniomaxillofacial surgery. Finally, there is the risk of bias, which was mitigated by clear inclusion/exclusion criteria and a dual review process to reduce error. This scoping review exclusively focused on publications in the academic realm that were identified via our literature review, as described in Section 2.3. As such, strictly commercial solutions were not included in this review.

#### 4.6.2. Limitations of AR and VR in Plastic and Craniomaxillofacial Surgery

Several limitations were noted throughout our search of the literature on VR/AR use in plastic and craniomaxillofacial surgery. There is a learning curve with respect to adopting the technology: Physician users expressed a need for training prior to AR use in metopic craniosynostosis repair [14]. Similarly, additional time needed to become familiar with the hardware and registration of AR holograms to patients slowed down operative times in orthognathic surgery [41]. This learning curve additionally extends to patients when AR/VR is used by them for pre-op planning (e.g., breast implant planning) [12]. Users with previous experience using VR/AR more easily implemented the system and had a more positive experience [12]. These users suggested that a technician can provide assistance with respect to the learning curve and any technical issues—although this could further contribute to greater costs [12]. 

There is also a dearth of readily available software for VR/AR use in plastic surgery. Many included studies had to develop in-house applications albeit based on existing software platforms. This may not be feasible for everyone attempting to implement these technologies into practice. Furthermore, software, depending on availability and technological capability, can be expensive and potentially increase operating costs. Although they were noted to be less expensive than some 3D printers, the development of a custom software platform by Cho et al. still costed approximately USD 16,500 and took 2 months to complete [27]. These challenges are reflected in the fact that all included studies presented platforms that were still in development rather than market-ready models.

The accuracy of hologram registration to patients was potentially decreased in low-light conditions, although the lighting was observed to be largely homogenous and a non-issue in the OR [14]. Several studies found that AR holograms struggled to accurately adapt to tissue deformation and depth intraoperatively and noted that it would be a key technological hurdle to overcome prior to further utilization [12,15,16,27]. In order to overcome the effects of deformation, one study described placing the patient in a prone position during imaging; furthermore, specific bony anatomic regions such as the legs were less impacted by the potential issue of tissue deformation [16]. Although time savings compared to 3D printing may be consequential, the relative novelty of AR technology makes investing in it inherently riskier as a result of greater technical requirements and less structural support availability [27]. Finally, there remains a deficit of well-constructed outcome studies examining VR/AR use within plastic surgery, particularly given the novelty of these technologies.

#### 4.6.3. Limitations of the Included Studies

A common limitation of many of the studies included in this review is a lack of quantitative comparative analysis between VR/AR methods and gold standard techniques and a lack of prospective controlled studies. Given this lack of quantitative data, we were unable to complete a systematic review or meta-analysis and instead structured this as a scoping review. This is primarily a result of the included studies presenting “proof of concept” workflows of their respective VR/AR platform. Although some of these platforms appear to be closer to widespread use than others, each one identified the need for further innovation and the quantification of their VR/AR tool’s benefits compared to current gold standards. The dearth of information quantifying VR/AR’s outcomes vs. traditional models was a key reason for this review to focus on what information was available and assess the current scope of VR/AR in its role as a clinician-as-user, patient-specific tool in plastic surgery. By determining the current scope of VR and AR in this function, aspiring adopters of similar technology might be better informed on current capacities.

## 5. Conclusions

The studies included in this review display the current capability of AR and VR technology. These capabilities include allowing unprecedented access to the patients’ own anatomy for the purpose of preoperative or intraoperative planning and execution, synergizing with existing 3D printing and potentially saving time during operations. However, more extensive outcome studies involving larger patient numbers are still required. This scoping review highlights the fact that there are limitations in our current application of these technologies that need to be improved in order for AR and VR to become more widely integrated into craniomaxillofacial surgery practice. Plastic and craniomaxillofacial surgeons have begun to introduce novel solutions that allow these technologies to be implemented in a clinician-as-user, patient-specific manner. The further development of VR and AR technology, coupled with ongoing innovations on the part of surgeons, engineers, and software developers, will determine the role that these tools will play in plastic surgery in the years to come.

## Figures and Tables

**Figure 1 bioengineering-10-00480-f001:**
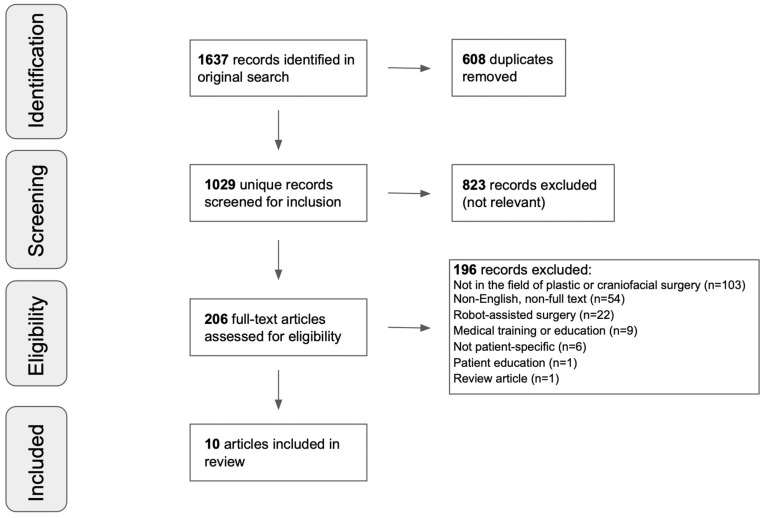
Literature review search results.

**Table 1 bioengineering-10-00480-t001:** Included studies and their characteristics.

Authors	Year	Institution	Publication Type	Surgical Stage	Procedure	Device	Software	Aim	Conclusion
Waked	2022	Department of Plastic and Reconstructive Surgery, University Hospital Brussel, Brussels, Belgium	Workflow description	Intraoperative	Facial artery mapping	Smartphone	Not reported	Test application and determine the accuracy of an AR app for visualizing patient-specific facial arterial anatomy	AR tool accurately visualized patient facial arterial anatomy and can contribute to safer dermal injections
Cho	2021	Department of Plastic Surgery, Cleveland Clinic	Workflow description	Preoperative	Facial vascularized composite allotransplantation	HoloLens	Unity platform and Visual Studio	Describe an AR workflow for use in facial transplantation	AR proved to be a time and cost saver, with the potential for pre- and intraoperative use
Garcia-Mato	2021	Departamento de Bioingenieria E Ingenieria Aerospacial, Universidad Carlos III De Madrid	Workflow description	Intraoperative	Metopic Craniosynostosis	Smartphone	Custom AR application based on the Unity platform (version 2019.3)	Develop AR preoperative planning method for craniosynostosis repair	AR workflow was successful both in practice and in real cases for assisting craniosynostosis repair
Mespreuve	2021	Department of Plastic and Reconstructive Surgery, University Hospital Brussel	Workflow description	Intraoperative	Dermal filler injection	Smartphone	Not reported	Examine the viability of pairing magnetic resonance angiography (MRA) with AR to visualize facial anatomy and avoid filler injection injuries	MRA and AR dual workflow was largely successful in identifying facial vasculature, and the proof of concept was successful
Coelho	2020	Santa Marcelina Hospital	Workflow description	Preoperative	Metopic Craniosynostosis	Smartphone	Custom AR application built using Unity framework and ARCore	Develop AR preoperative planning method for craniosynostosis repair	Their AR workflow can be used to visualize patient-specific anatomy
Wesselius	2020	Department of Oral and Maxillofacial Surgery, Radboud University Medical Center	Workflow description	Intraoperative	Deep inferior epigastric perforator (DIEP) flap	HoloLens	In-house developed HoloLens application (using Unity framework)	Describe an AR workflow designed to visualize vessels for a DIEP flap	Their AR workflow can be used to visualize patient-specific anatomy
Yaremenko	2020	Department of Maxillofacial Surgery, Pavlov University	Case report and workflow description	Intraoperative	Auricular reconstruction	Epson Moverio BT-300	Not reported	Examine the use of AR for microtia correction	AR was useful in visualizing anatomy and conducting a microtia correction
Amini	2019	Department of Computer Science and Software Engineering, Concordia University	Workflow description	Preoperative	Single Mastectomy	HoloLens	Custom AR application built using Unity version 2018.1.0 and the Vuforia SDK Engine.	Present an augmented reality application, which enables surgeons to see the shape of the implants, as 3D holograms on the patient’s body.	AR can be used to model 3D objects in real time with some subject education
Giacalone	2019	Department of Lymphatic Surgery, Sint-Maarten Hospital	Workflow description and Case report	Preoperative	Lymphovenous Anastomosis	Medicalholodeck	Custom VR application using Medicalholodeck platform and software	Describe a VR workflow designed to preoperatively plan a lymphatic malformation repair	VR workflow was successful in visualizing complex anatomy in streamlining the operation
Pratt	2018	Department of Surgery and Cancer, Imperial College London	Case series and workflow description	Intraoperative	Deep inferior epigastric perforator (DIEP) flap	HoloLens	Custom AR application based on the Unity platform (version 2017.1)	Describe an AR workflow designed to visualize vessels for a DIEP flap and demonstrate accuracy in patients	AR workflow was demonstrated to be accurate in viewing patient anatomy and assisting in DIEP procedures

## Data Availability

Not applicable.

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
