# Peer review of "Virtual Reality and Augmented Reality in Plastic and Craniomaxillofacial Surgery: A Scoping Review"

_bioengineering, 2023, doi:10.3390/bioengineering10040480_

Round 1

Reviewer 1 Report

The technology of virtual reality (VR) and augmented reality (AR) have evolved has been developed rapidly in the past three decades. More powerful software and miniaturized hardware were employed in surgical practice. In this work, the authors reviewed the VR/AR in plastic and craniomaxillofacial surgery. I think the manuscript can be published in Bioengineering after revision.

Comments:

1.      It seems that this review lacks of quantitative comparative analysis between VR/AR methods. I think the authors can list some data about the comparison.

2.      What is the cost & time consumption for the VR/AR instrument used in the surgery now? and its variation during the recent years. If possible, the authors can give a graph to describe the changes of the cost the time for AR/VR instruments.

3.      What is the reason for the error during utilization of AR/VR in plastic and craniomaxillofacial surgery, and is there and methods to reduce it?

4.    In Conclusions, the authors can give their own prospective for the AR/VR technology in the future plastic and craniomaxillofacial surgery.

Author Response

Point 1:

 It seems that this review lacks of quantitative comparative analysis between VR/AR methods. I think the authors can list some data about the comparison.

Response 1:

Regarding the dearth of quantitative comparison between VR and AR methodologies, we agree that there is a need for further elaboration. To a certain extent, we have been limited by a lack of discussion of this subject by the included studies themselves. However, we have returned to the discussion section where we have added context under a new section, ‘Methodologies of the Included Studies,’ which reads as follows:
“A variety of methodologies were implemented among the included studies, with an overarching pattern of the use of a VR or AR platform in a case or series of cases. Regardless of the procedure of use and time of implementation, 3D imaging was initially conducted for transference to AR/VR platforms. The only exception to this was Amini et al, who did not need to visualize participant anatomy as they instead projected the 3D presence of a breast implant for mastectomy reconstruction patients12. The majority of  studies paired their VR or AR application with a model;  Cho et al, Yaremenko et al, Coelho et al, and Garcia-Mato et al utilized 3D printed, patient-specific models parallel to their virtual platform13,14,28,31. The accuracy of the virtual platforms was validated to varying degrees. Cho et al compared their AR hologram to 3D CT data, while Amini et al compared theirs to an inflatable model12,28. Garcia-Mato overlaid their AR model onto a 3D printed model14. Ultrasound was used to validate accuracy in studies wherein AR was used to identify vasculature16,30. In the single VR study, preoperative planning of lymphaticovenous anastomosis was checked intraoperatively using indocyanine green lymphography15. Notably, while AR platforms can be utilized either pre- or intraoperatively, VR applications are primarily used as planning tools preoperatively.”

Point 2:

What is the cost & time consumption for the VR/AR instrument used in the surgery now? and its variation during the recent years. If possible, the authors can give a graph to describe the changes of the cost the time for AR/VR instruments.

Response 2:

Regarding the role of AR/VR as cost and time-saving tools both prior and during operations, we agree that there is a need for further quantification of the savings enabled by implementing these technologies. In fact, this was a subject we had hoped to emphasize when this study was conceptualized. However, the articles in the scope of our review largely do not discuss the subject, as they consist of workflow studies. 

Point 3:

What is the reason for the error during utilization of AR/VR in plastic and craniomaxillofacial surgery, and is there and methods to reduce it?

Response 3:

This is a question we had intended to address as part of this scoping review. However, among the ten included articles, there was no specific description of this topic. The single encompassing source of error we identified was that of projections adapting to tissue deformation which we cited in our ‘limitations’ section: “Several studies found that AR holograms struggled to accurately adapt to tissue deformation and depth intraoperatively and noted that it would be a key technological hurdle to overcome prior to further utilization”. We have added further context towards this point in the limitations section. 

Point 4:

In Conclusions, the authors can give their own prospective for the AR/VR technology in the future plastic and craniomaxillofacial surgery.

Response 4:

We thank the reviewer for their request to see our perspective on the role of AR/VR in the future of plastic and craniomaxillofacial surgery. We believe we did address this matter in our conclusion section: “more extensive outcome studies involving larger patient numbers are still required,” “limitations … need to be improved in order for AR and VR to become more widely integrated into craniomaxillofacial surgery practice,” “further development of VR & AR technology, coupled with ongoing innovation on the part of surgeons, engineers and software developers, will determine the role which these tools will play in plastic surgery in years to come”. In short, our understanding is that–while promising–further technical advancements and quantitative research will be necessitated before we can better understand the extent to which VR and AR can be integrated into plastic and craniomaxillofacial surgery.

Reviewer 2 Report

Comments on Kaplan et al:

The aim of this scoping review is to perform a complete analysis of literature, by including all articles (2018 – 2021), about VR and AR application in plastic and craniofacial surgery, in a clinician- as- user, patient – specific – manner.

This manuscript shows rich content, providing a deep insight for some works: the study is within the journal’s scope, and I found it to be well-written, providing sufficient information. Even if the manuscript provides an organic overview, with a densely organized structure and based on well-synthetized evidence, there are some suggestions necessary to make the article complete and fully readable. For these reasons, the manuscript requires major changes.

Please find below an enumerated list of comments on my review of the manuscript:

INTRODUCTION:

LINE 31: Virtual dissection table, which is an interactive device, that provide male and female cadaver datasets, by different modes (grass anatomy and high resolution methods), is considered a promising key element in the anatomical teaching (see, for reference: Bianchi, S.; Bernardi, S.; Perilli, E.; Cipollone, C.; Di Biasi, J.; Macchiarelli, G. Evaluation of Effectiveness of Digital Technologies During Anatomy Learning in Nursing School. Appl. Sci. 202010, 2357. https://doi.org/10.3390/app10072357). The authors should highlight the key role, played by forefront and virtual devices in anatomical teaching.

LINE 35: Please, reformulate this sentence as following “These technologies primarily strives to bridge the gap between two-dimensional imaging modalities and the three-dimensional nature of surgical procedures”.

LINE 37: There is an article usage problem in this sentence. Rewrite the sentence as following: “ Computer-aided design software allows for the manipulation of  the patient anatomical data, yet is limited in usability by its technical complexity and often-prohibitive cost”.

LINE 89: Furthermore, multimodal imaging is required to formulate a differential diagnosis in tumors surgery applications, or vascular malformations, by providing diagnostic evidence on soft tissues (see, for reference: Somma L, Iacoangeli M, Nasi D, Balercia P, Lupi E, Girotto R, Polonara G, Scerrati M. Combined supra-transorbital keyhole approach for treatment of delayed intraorbital encephalocele: A minimally invasive approach for an unusual complication of decompressive craniectomy. Surg Neurol Int. 2016 Jan 7;7(Suppl 1):S12-6. doi: 10.4103/2152-7806.173561. PMID: 26862452; PMCID: PMC4722521).

The main topic is interesting, and certainly of great clinical impact. As regards the originality and strengths of this manuscript, this is a significant contribute to the ongoing research on this topic, as it extends the research field on the VR and AR application in plastic and craniofacial surgery. Overall, the contents are rich, and the authors also give their deep insight for some works.

As regards the section of methods, there is a specific and detailed explanation for the methods used in this study: this is particularly significant, since the manuscript relies on a multitude of methodological and statistical analysis, to derive its conclusions. The methodology applied is overall correct, the results are reliable and adequately discussed.

The conclusion of this manuscript is perfectly in line with the main purpose of the paper: the authors have designed and conducted the study properly. As regards the conclusions, they are well written and present an adequate balance between the description of previous findings and the results presented by the authors.

Finally, this manuscript also shows a basic structure, properly divided and looks like very informative on this topic. Furthermore, figures and tables are complete, organized in an organic manner and easy to read.

In conclusion, this manuscript is densely presented and well organized, based on well-synthetized evidence. The authors were lucid in their style of writing, making it easy to read and understand the message, portrayed in the manuscript. Besides, the methodology design was appropriately implemented within the study. However, many of the topics are very concisely covered. This manuscript provided a comprehensive analysis of current knowledge in this field. Moreover, this research has futuristic importance and could be potential for future research. However, major concerns of this manuscript are with the introductive section: for these reasons, I have major comments for this section, for improvement before acceptance for publication. The article is accurate and provides relevant information on the topic and I have some major points to make, that may help to improve the quality of the current manuscript and maximize its scientific impact. I would accept this manuscript if the comments are addressed properly.

Author Response

Point 1:

LINE 31: Virtual dissection table, which is an interactive device, that provide male and female cadaver datasets, by different modes (grass anatomy and high resolution methods), is considered a promising key element in the anatomical teaching (see, for reference: Bianchi, S.; Bernardi, S.; Perilli, E.; Cipollone, C.; Di Biasi, J.; Macchiarelli, G. Evaluation of Effectiveness of Digital Technologies During Anatomy Learning in Nursing School. Appl. Sci. 2020, 10, 2357. https://doi.org/10.3390/app10072357). The authors should highlight the key role played by forefront and virtual devices in anatomical teaching.

Response 1:

Thank you for the suggestion, we have made this modification and added it into the introduction section to provide context as to the multifaceted use of AR/VR in the medical setting.

Point 2:

LINE 35: Please, reformulate this sentence as following “These technologies primarily strives to bridge the gap between two-dimensional imaging modalities and the three-dimensional nature of surgical procedures”.

Response 2:

Line 35: Thank you for the suggestion, we have made this modification.

Point 3:

LINE 37: There is an article usage problem in this sentence. Rewrite the sentence as following: “ Computer-aided design software allows for the manipulation of  the patient anatomical data, yet is limited in usability by its technical complexity and often-prohibitive cost”.

Response 3:

Line 37: Thank you for the suggestion, we have made this modification.

Point 4:

LINE 89: Furthermore, multimodal imaging is required to formulate a differential diagnosis in tumors surgery applications, or vascular malformations, by providing diagnostic evidence on soft tissues (see, for reference: Somma L, Iacoangeli M, Nasi D, Balercia P, Lupi E, Girotto R, Polonara G, Scerrati M. Combined supra-transorbital keyhole approach for treatment of delayed intraorbital encephalocele: A minimally invasive approach for an unusual complication of decompressive craniectomy. Surg Neurol Int. 2016 Jan 7;7(Suppl 1):S12-6. doi: 10.4103/2152-7806.173561. PMID: 26862452; PMCID: PMC4722521).

Response 4:

Line 89: Thank you for the suggestion, we have made this modification. It has been added in the first paragraph for continuity.

Reviewer 3 Report

This work aims to review the VR and AR applied in plastic and craniofacial surgeons. A few issues need to be modified before acceptance. There are different formats in the article, such as 2.1 and 2.2. And more charts should be summarized and analyzed in the paper.

Author Response

Point 1:

This work aims to review the VR and AR applied in plastic and craniofacial surgeons. A few issues need to be modified before acceptance. There are different formats in the article, such as 2.1 and 2.2. And more charts should be summarized and analyzed in the paper.

Response 1:

Thank you for pointing out the formatting error in section 2.1. We have corrected it along with reviewing the overall manuscript and making changes to ensure a consistent format. We also carefully considered your recommendation to include more charts. With only ten articles selected in this scoping review, we are afraid that additional graphics based upon such limited data may misrepresent the current scope of this field and thus mislead readers.

Reviewer 4 Report

This paper gives a review about AR/VR use in various surgical applications focusing on plastic and cranio-maxillofacial interventions. Authors conclude that AR/VR is helpful and beneficial, but there are too few results and further use cases are needed.

Although the paper is scientifically correct, in my opinion, it contributes only marginally to the area and brings nothing new to the reader. 

My recommendation is rejection and re-submission to a proper conference as a conference paper (after shortening).

Some detailed comments:

Overall merit: the applied scientific method for selection and comparison is correct and described deeply. Authors spent a lot of time and work to pre-select papers. On the other hand, most part of the paper describes the procedure itself, how to restrict the number of candidate papers from 1600+ to only ten. Ten papers are simply too few. Later, this ten papers are described, summarized and compared. 

Basically, for me, reading the abstract and the conclusion would have been enough. The conclusion, that using AR/VR is advantageous, but still unexplored deeply is a correct statement and outcome, but not really new. The big table is helpful for doctors who are interested in facial surgery with AR/VR. It is quite limited. I think this review is not very interesting for a wider audience in general bioengineering, and I would therefore redirect it after major shortening to a "facial-plastic" conference.

 I don't know if it is allowed to use the & sign in a title. Even if it is, I would use the "and" word instead of the &.

 Time saving issues: I think this should be more clearly addressed. The use of AR/VR during operation as a time saving factor is very important, because the time spent in the operation room is more valuable (more expensive) than the time spent with pre-operation modelling and practice. Furthermore, the increasing probability of success is even more important. However, calculating the overall time spent for the procedure, including creation 3D models, processing, printing etc. is an important issue. It is hard to compare, but the "hourly rate" of a surgeon in the operation room is much more than it is for a "3D assistant ". This kind of evaluation would do good for the paper to enforce the use of AR/VR. 

Moreover, the role of training and education could be evaluated deeper and highlighted. We have also experience with the Hololens (and it was found to big and not ergonomic enough to use it the operation room), furthermore, also with 3D printing of individual body models (tissue, bones, vessels), together with AR-projected images. Mostly for cardiac surgery, but also for reconstruction of ears, noses and lately for implanting a printed zygomatic bone of titanium. It is used for personalized interventions as pre-surgery training and post-surgery demonstrations for the students. (It is based on CT and/or real scans of the body). One of my major concern was with this paper that it should be widened, beyond only 10 scientific papers, and maybe look around in the "business area", as there are some commercial solutions available on the market. I do not have myself any business-based conflict of interest, but I still didn't want to suggest commercial websites we use to order 3D prints for facial surgery. 

Author Response

Point 1: 

Overall merit: the applied scientific method for selection and comparison is correct and described deeply. Authors spent a lot of time and work to pre-select papers. On the other hand, most part of the paper describes the procedure itself, how to restrict the number of candidate papers from 1600+ to only ten. Ten papers are simply too few. Later, this ten papers are described, summarized and compared. 

Response 1:

We elected to conduct a scoping review, a relatively novel study design alternative to systematic reviews in clinical research. Munn et al 2018 describes the distinction between these 2 designs, commenting that the role of a scoping review is to “identify knowledge gaps, scope of a body of literature, clarify concepts or to investigate research conduct.” Despite the relative novelty of this methodology, we believe it is an appropriate design in this interdisciplinary study between medicine and engineering. We acknowledge this reviewer’s concerns regarding the number of studies selected for this review. However, the relatively narrow scope we selected in this crossover between plastic and craniomaxillofacial surgery and VR/AR technology precludes a larger sample size. Our focus on clinician-as-user, patient-specific use of VR and AR–while critical to tailoring this study to clinicians who are interested in understanding how they can directly apply these technologies towards the treatment of their patients–limits our discussion of other applications of VR and AR in the clinical setting. Because of this, we detailed the methodology which took us from the original search to just 10 articles. We also attempted to dedicate all the more focus on the specifics of the 10 included papers because we focused on so few studies. We agree that AR/VR use as training and education tools is a major strength of these platforms; yet, as this falls outside the scope of our inclusion criteria, we elected to limit our discussion of these aspects of use. Nonetheless, we did elaborate with an example in the introduction section: “For instance, the development of virtual dissection tables, which are interactive devices that provide male and female cadaver datasets by different modes (gross anatomy and high resolution methods), is a promising key element in anatomical teaching21

Reference:

Munn Z, Peters MDJ, Stern C, Tufanaru C, McArthur A, Aromataris E. Systematic review or scoping review? Guidance for authors when choosing between a systematic or scoping review approach. BMC Med Res Methodol. 2018 Nov 19;18(1):143. doi: 10.1186/s12874-018-0611-x. PMID: 30453902; PMCID: PMC6245623.

Point 2:

 I don't know if it is allowed to use the & sign in a title. Even if it is, I would use the "and" word instead of the &.

Response 2:

Regarding the use of the ampersand in our title, thank you for pointing it out. We have corrected it accordingly.

Point 3:

 Time saving issues: I think this should be more clearly addressed. The use of AR/VR during operation as a time saving factor is very important, because the time spent in the operation room is more valuable (more expensive) than the time spent with pre-operation modelling and practice. Furthermore, the increasing probability of success is even more important. However, calculating the overall time spent for the procedure, including creation 3D models, processing, printing etc. is an important issue. It is hard to compare, but the "hourly rate" of a surgeon in the operation room is much more than it is for a "3D assistant ". This kind of evaluation would do good for the paper to enforce the use of AR/VR. 

Response 3:

We agree that there is a need for further quantification of the savings enabled by implementing these technologies. In fact, this was a subject we had hoped to emphasize when this study was conceptualized. However, the articles in the scope of our review largely do not discuss the subject, as they consist of workflow studies. Any material within the selected studies that pertained to time-savings was discussed in as great of detail as possible; yet, the studies themselves emphasized a need for additional investigation of how VR and AR platforms affect the time efficiency of operating. Our review of these 10 articles is in agreement with the reviewer’s assessment that further study is needed in this aspect of VR and AR platforms. 

Point 4:

Moreover, the role of training and education could be evaluated deeper and highlighted. We have also experience with the Hololens (and it was found to big and not ergonomic enough to use it the operation room), furthermore, also with 3D printing of individual body models (tissue, bones, vessels), together with AR-projected images. Mostly for cardiac surgery, but also for reconstruction of ears, noses and lately for implanting a printed zygomatic bone of titanium. It is used for personalized interventions as pre-surgery training and post-surgery demonstrations for the students. (It is based on CT and/or real scans of the body). 

Response 4:

Thank you for sharing your experience with the HoloLens. We have now addressed this issue in the discussion section as follows: “Moreover, the same digital rendering and AR hologram may be transferable between various hardware platforms. For example, when users found the HoloLens headset to be cumbersome in Amini et al, authors suggested shifting the subsequent iteration of their platform to a smartphone application.”  

Response 5:

One of my major concern was with this paper that it should be widened, beyond only 10 scientific papers, and maybe look around in the "business area", as there are some commercial solutions available on the market. I do not have myself any business-based conflict of interest, but I still didn't want to suggest commercial websites we use to order 3D prints for facial surgery. 

Point 5:

This scoping review focused on publications in the academic realm using sources from Pubmed. As such, we have not yet reviewed strictly commercial solutions.

Round 2

Reviewer 2 Report

Manuscript can be now accepted

Author Response

Thank you!

Reviewer 4 Report

Authors revised their paper according to the suggestions of the reviewers. Based on their response, some comments follow.

Some minor suggestions:

Although the authors highlighted the difference between systematic and scoping review and they use the term throughout the paper, furthermore, we can assume that readers are familiar with this, I would include one or two sentences about this with the reference Munn they provided in the response. Just to make it clear (already in the introduction and at the motivation) could be helpful.

Section 4.5

It is very good to point out time saving factors. We are here in the discussion section, so there is room for opinion of the authors. I also found their response here good enough to place it (1-2 sentences) also in the paper. Where and how can we save money (and time), that we need more data and evaluation how to do this and that the papers reviewed did not deal with this issue deeply.

The same is true for response 5. It is clear and well done to cover scientific papers only. There are many purely commercial solutions (usually companies run by doctors). Although I would consider to dedicate a short section for this, I just recommend the authors reconsider to mention (even without any references to actual commercial websites) the fact that there exist commercial solutions. They are the proof of that this technology is "time and cost saving", and if readers are done with the paper they may look after what is available already as service. 

I suggest to check out these references as well, not necessarily related to facial issues, but handling the problems of costs, time and business in general for 3D printing.

Ventola, C. L. (2014). Medical applications for 3D printing: current and projected uses. Pharmacy and Therapeutics39(10), 704.

Javaid, M., Haleem, A., Singh, R. P., & Suman, R. (2022). 3D printing applications for healthcare research and development. Global Health Journal.

Chen, J. K., & Do, H. T. (2017, July). Perspective of the 3D Printing Technology Applied on Medical Resource Integration and Service Innovation Business Model. In 2017 Portland International Conference on Management of Engineering and Technology (PICMET) (pp. 1-11). IEEE.

Author Response

Point 1:

Although the authors highlighted the difference between systematic and scoping review and they use the term throughout the paper, furthermore, we can assume that readers are familiar with this, I would include one or two sentences about this with the reference Munn they provided in the response. Just to make it clear (already in the introduction and at the motivation) could be helpful.

Response 1: 

Thank you for your suggestion. We modified accordingly in section 2.1. It appears as follows:

“This is a scoping review, a relatively novel study design alternative to systematic reviews in clinical research. As Munn et al 2018 described, a scoping review is to “identify knowledge gaps, scope of a body of literature, clarify concepts or to investigate research conduct.”

Point 2:

It is very good to point out time saving factors. We are here in the discussion section, so there is room for opinion of the authors. I also found their response here good enough to place it (1-2 sentences) also in the paper. Where and how can we save money (and time), that we need more data and evaluation how to do this and that the papers reviewed did not deal with this issue deeply.

Response 2:

Thank you for your suggestion. We added a few sentences in section 4.5. It appears as follows:

“Overall, there remains a need for further quantification of the savings enabled by implementing these technologies. The majority of the articles included in this review primarily focused on the development and implementation of novel workflows using these technologies. As a result, they did not did not comprehensively delineate nor compare the effect of their platform on surgical costs and timing. There is a need for further evaluation of these variables.”

Point 3:

The same is true for response 5. It is clear and well done to cover scientific papers only. There are many purely commercial solutions (usually companies run by doctors). Although I would consider to dedicate a short section for this, I just recommend the authors reconsider to mention (even without any references to actual commercial websites) the fact that there exist commercial solutions. They are the proof of that this technology is "time and cost saving", and if readers are done with the paper they may look after what is available already as service. 

Response 3:

This is an academic review article so we do not cite commercial products. We briefly explained this as you suggest in section 4.6.1. As follows:

"This scoping review exclusively focused on publications in the academic realm which were identified via our literature review, as described in Methods 2.3. As such, strictly commercial solutions were not included in this review."

Point 4:

I suggest to check out these references as well, not necessarily related to facial issues, but handling the problems of costs, time and business in general for 3D printing.

Response 4:

Thank you for suggesting these articles relevant to the commercial use of 3D printing in medicine. While we find that these articles are relevant to the discussion of 3DP and effects on cost and timing in medicine in general, we believe they fall outside the scope of our review which focuses specifically on AR and VR. We did mention 3DP in the introduction/background and hope that this addresses your helpful suggestions.